# Real-time forecasting of COVID-19-related hospital strain in France using a non-Markovian mechanistic model

Alexander Massey[1][☺]*, Corentin Boennec[2][☺], Claudia Ximena Restrepo-Ortiz[3], Christophe Blanchet[4], Samuel Alizon[1,5][‡], Mircea T. Sofonea[1,6,7][‡]

**1** Infectious Diseases and Vectors: Ecology, Genetics, Evolution and Control (MIVEGEC), Université de Montpellier, National Centre for Scientific Research (CNRS), French National Research Institute for Sustainable Development (IRD), Montpellier, France, **2** Laboratoire Plasma et Conversion d'Energie (LAPLACE), National Centre for Scientific Research (CNRS), Institut National Polytechnique de Toulouse (Toulouse INP), Université Toulouse 3-Paul Sabatier, Toulouse, France, **3** MARine Biodiversity, Exploitation & Conservation (MARBEC), Université de Montpellier, National Centre for Scientific Research (CNRS), French National Institute for Ocean Science and Technology (Ifremer), French National Research Institute for Sustainable Development (IRD), Montpellier, France, **4** Institut Français de Bioinformatique, IFB-core UAR 3601, National Centre for Scientific Research (CNRS), Évry, France, **5** Center for Interdisciplinary Research in Biology (CIRB), Collège de France, National Centre for Scientific Research (CNRS), National Institute of Health and Medical Research (Inserm), Université Paris Sciences et Lettres, Paris, France, **6** Pathogenesis and Control of Chronic and Emerging Infections (PCCEI), Université de Montpellier, National Institute of Health and Medical Research (Inserm), Établissement Français du Sang (EFS), Université des Antilles, Montpellier, France, **7** Centre Hospitalier Universitaire de Nîmes (CHU de Nîmes), Nîmes, France

☺ These authors contributed equally to this work.
‡ These senior authors also contributed equally to this work.
* stats.massey@gmail.com

**Data Availability Statement:** The model in this manuscript is based on publicly available data provided by Santé Publique France (https://www.data.gouv.fr/fr/datasets/). The scripts and data

## Abstract

Projects such as the European Covid-19 Forecast Hub publish forecasts on the national level for new deaths, new cases, and hospital admissions, but not direct measurements of hospital strain like critical care bed occupancy at the sub-national level, which is of particular interest to health professionals for planning purposes. We present a sub-national French framework for forecasting hospital strain based on a non-Markovian compartmental model, its associated online visualisation tool and a retrospective evaluation of the real-time forecasts it provided from January to December 2021 by comparing to three baselines derived from standard statistical forecasting methods (a naive model, auto-regression, and an ensemble of exponential smoothing and ARIMA). In terms of median absolute error for forecasting critical care unit occupancy at the two-week horizon, our model only outperformed the naive baseline for 4 out of 14 geographical units and underperformed compared to the ensemble baseline for 5 of them at the 90% confidence level ($n = 38$). However, for the same level at the 4 week horizon, our model was never statistically outperformed for any unit despite outperforming the baselines 10 times spanning 7 out of 14 geographical units. This implies modest forecasting utility for longer horizons which may justify the application of non-Markovian compartmental models in the context of hospital-strain surveillance for future pandemics.

used to perform the analysis and generate this manuscript are available on GitLab (https://gitlab.in2p3.fr/ete/covidici_public) and archived in Zenodo (doi:10.5281/zenodo.7641132). The companion web dashboard is hosted by France Bioinformatique (https://cloudapps.france-bioinformatique.fr/covidici/).

**Funding:** Centre National de la Recherche Scientifique, MODCOVD19/INSMI PaSSES project to AM; Université de Montpellier to AM and CXRO; Région Occitanie, ANR Flash PHYEPI project to CB; Université de Montpellier to MTS; Centre National de la Recherche Scientifique to SA; Agence Nationale de la Recherche, ANR-11-INBS-0013 to CB. The funders had no role in study design, data collection and analysis, decision to publish, or preparation of the manuscript.

**Competing interests:** The authors have declared that no competing interests exist.

## Author summary

The US and European Covid-19 Forecast Hubs focus on metrics such as deaths, new cases, and hospital admissions, but do not offer measurements of hospital strain like critical care bed occupancy, which was essential for the provisioning of healthcare resources during the COVID-19 pandemic. Furthermore, forecasting support was only guaranteed on the national level leaving many countries to look elsewhere for valuable sub-national forecasts. In France statistical modelling approaches were proposed to anticipate hospital stain at the sub-national level but these were limited by a two-week forecast horizon. We present a sub-national French modelling framework and online application for anticipating hospital strain at the four-week horizon that can account for abrupt changes in key epidemiological parameters. It was the only publicly available real-time non-Markovian mechanistic model for the French epidemic when implemented in January 2021 and, to our knowledge, it still was at the time it stopped in early 2022. Further adaptations of this surveillance system can serve as an anticipation tool for hospital strain across sub-national localities to aid in the prevention of short-noticed ward closures and patient transfers.

## Introduction

The COVID-19 pandemic emphasised that policymakers need access to accurate forecasts of key epidemiological indicators to mitigate strain on hospital services and reduce preventable deaths [1, 2]. This has led to international projects tasked with creating centralised repositories for COVID-19 forecasts pertaining to the United States [3], Germany/Poland [4], and Europe [5]. Sub-national forecasts were of particular interest as policies implemented at the local level tend to outperform uniform national policies [6] and as the geographic resolution of the forecasts is a relevant factor for their effectiveness [7]. However, in contrast to its counterparts, the European COVID-19 Forecast Hub, as well as many other COVID-19 forecasting analyses [8, 9], only considers the national level. This left countries like France to rely on other sources to optimise their local to national healthcare system management.

Of particular interest is intensive care unit (ICU) occupancy, one of the most direct indicators of hospital strain, which is commonly predicted using either statistical or mechanistic models. Statistical models use correlations in previously observed data to explain model structure that can be separated from noise and extrapolated to the future. In time series analysis, they can benefit from the addition of adequate predictor variables. For example, in France [10] proposed an ensemble model approach that combined several statistical models (including machine learning) to utilise predictors identified from available epidemiological, mobility, and meteorological data to make 14-day forecasts for ICU admissions and ICU occupancy by French region. Their ensemble was effective in the short-term but had more difficulty predicting beyond the lag (typically at most two weeks) between their predictors (e.g. positive antigen testing) and the subsequent hospitalisation events.

Mechanistic models are explicitly based on a simplified version of the underlying epidemiological process [11]. The most popular is the compartmental model, which typically involves separating a population into distinct sub-populations (e.g. susceptible, infected, and "removed" individuals in the SIR model) and inferring the transition dynamics between these compartments. This can be done based on assumptions regarding the biology of the pathogen or via optimisation approaches using observed data, e.g. hospital admissions, to make inferences regarding partially or unobserved factors such as daily infections [12]. The biological

assumptions simplify the causal relationships between a pathogen's infectivity, pathogenicity, and lethality so that long-term forecasts can be produced. These models provide us with a mechanistic understanding of the epidemic process and can help to anticipate planned changes such as lockdowns or increases in vaccination coverage. A limitation of compartmental models is that they typically require large, idealised populations for best results [13] and can have lower predictive performance [14] depending on the time scale. However, as shown in [15], a non-Markovian discrete-time compartmental model may have the potential to capture the dynamics up to 5 weeks on average (although this also reflects the epidemiological relevance of the underlying assumptions made).

To facilitate COVID-19 monitoring in France, we developed `COVIDici`: a mechanistic transmission model that accurately captures the hospital and mortality dynamics from the French epidemic. Model results were publicly communicated via a web dashboard (https://cloudapps.france-bioinformatique.fr/covidici/), which provided real-time visualisations of the French epidemic at the national, regional, and departmental levels. `COVIDici` was updated daily using databases published by the national public health agency [16] until 2022 when it was halted after the emergence of the Omicron variant [17] led to decreased interest in epidemic forecasting by French authorities, partly driven by the belief that Omicron BA.1 would represent our way out of the pandemic and the last wave [18].

Here, we briefly summarise the structure of the underlying compartmental model, the statistical procedure for the parameter inference and describe communication via the web dashboard. We present an exploratory data analysis to retrospectively evaluate `COVIDici`'s forecasts for ICU occupancy up to the four-week horizon at the regional and national levels using standard metrics for continuous variables as well as a binarised version representing ICU overload to focus on model performance in anticipating wave peaks. Standard statistical forecasting methods (auto-regression, exponential smoothing, and ARIMA) are included as baseline models. Finally, we discuss perspectives for COVID-19 epidemic modelling in the context of decreased surveillance and what this means for the surveillance of pandemics in the future.

## Materials and methods

This section presents a summary of `COVIDici` and its retrospective evaluation. `COVIDici` is the sub-national extension on a pre-existing discrete-time epidemiological model (`COVIDSIM-FR`, [19]), developed in `R` [20], that combines the computational benefits of deterministic dynamical systems with the short-time accuracy of non-Markovian dynamics. `COVIDSIM-FR` was initially tailored to capture the dynamics of national ICU bed occupancy during the first COVID-19 wave in France (March-May 2020) [21], and `COVIDici` was further developed to pursue this objective at sub-national levels and taking into account both the vaccination campaign and viral evolution.

The scripts and data used to perform the analysis and generate this manuscript are available on GitLab (https://gitlab.in2p3.fr/ete/covidici_public) and archived in Zenodo [22]. The current section provides a overview of the main methods used in both the implementation of `COVIDici`, while further technical details regarding the implementation of `COVIDici` as well as all the baseline models is accessible in S1 Text. The mathematical and computational foundations of the model behind `COVIDici` are exposed in [21].

### The model

The structure of the model, shown in Fig 1, describes the flows between the relevant clinical-epidemiological compartments within hospitals as well as the greater community. It includes

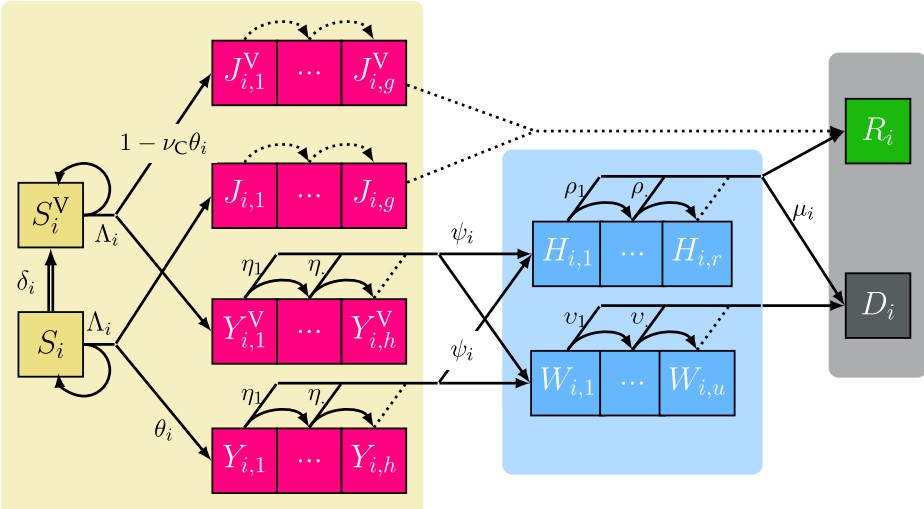

**Fig 1. Structure of the underlying COVID-19 epidemic discrete time model with vaccination.** The three shaded areas represent the general population or community (left), the critically-ill hopitalised patients (center) and individuals removed by either recovery or death (right). The first subscript ($i$) of compartment densities (capital letters) indicates the age class, while the second denotes the time (in days) elapsed since the entry into the compartment—$g$, $h$, $r$, and $u$ are thus the maximum number of days possible to remain in a compartment represented by contiguous boxes (see [21] for their computation). Individuals in $S$ are susceptible, $J$ are non-critically infectious, $Y$ are infected and will eventually require hospitalisation, $H$ are hospitalised in a critical care bed for a long stay, $W$ are in non-ICU beds, $R$ are recovered and $D$ are deceased. $S^V$, $J^V$ and $Y^V$ are the vaccinated counterparts of $S$, $J$ and $Y$. $\Lambda_i$ is the daily force of infection (a probability). $\delta_i$ is the daily vaccination rate. $\mu_i$, $\psi_i$, and $\theta_i$ are transition rates between compartments, the latter being reduced by the factor $\nu_C$ for vaccinated individuals. Arrows between boxes show the daily flow of individuals between compartments where dotted arrows occur with probability 1. For the sake of simplicity, only one age group is depicted here and only one of the two complementary probabilities is shown for each bifurcating transition.

individuals that, by assumption [23], no longer contribute to the epidemic due to recovery with full immunity or death. Stratifying by age, where the index $i$ denotes the individual age class, each box represents a group of individuals who share the same clinical kinetics and who contribute equally to the epidemic dynamics. Most susceptible individuals in $S_i$ will pass to the non-critically ill compartment, denoted $J_i$, with rate proportional to $(1 − \theta)$. A small fraction $\theta$ of the infected individuals, which increases with age [24], reaches the critical infection compartment, $Y_i$, meaning they will eventually be hospitalised either for a long stay in critical care wards, $H_i$, or they will be in conventional hospital care ($W_i$) where they end up in the deceased compartment $D_i$ with probability 1. Critically-ill individuals move to $H_i$ with probability $\psi_i$ and might die with probability $\mu_i$. Those who do not die enter the recovered compartment $R_i$ where they are assumed to have perfect immunity for the rest of the simulation period. $\eta$, $\rho$ and $\nu$ reflect the daily probabilities to exit $Y_i$, $H_i$ and $W_i$ respectively.

Aged care facilities (i.e. nursing homes) are ignored in this model because of their differences in hospitalisation rates and epidemiological dynamics compared to the general population [25]. Importantly, individuals are stratified according to vaccination status (upon their first dose) prior to hospitalisation: the vaccinated compartments are denoted by the exponent $^V$. $\delta_i$ is the daily vaccination rate for age class $i$, given by, or linearly extrapolated according to the trend of the previous 3 weeks, from the French vaccination database VAC-SI [26].

The most important quantity of the `COVIDici` model is the daily rate at which susceptible hosts become infected, namely the discrete-time force of infection (FOI) $\Lambda_i(t)$. For a given

national or sub-national location $\ell$ and calendar date $t$, FOI is defined by

$$\Lambda_{i,\ell}(t) := \left(1 + \left(\frac{\mathcal{R}_0}{S_\ell^\circ}\mathcal{C}_\ell(t)\sum_{\tau=1}^{\tau_{\mathrm{Z}}^{\max}}\zeta_\tau\sum_{j\in\mathcal{A}}J_{j,\tau}(t) + Y_{j,\tau}(t) + v_{\mathrm{T}}\left(J_{j,\tau}^{\mathrm{V}}(t) + Y_{j,\tau}^{\mathrm{V}}(t)\right)\right)^{-1}\right)^{-1}, \qquad (1)$$

where $\mathcal{R}_0$ is the basic reproduction number of the original (Wuhan) strain of SARS-CoV-2 in France, $S^\circ$ is the population size in location $\ell$, and $C_\ell(t)$ is the piecewise-constant transmission factor that captures all spatiotemporal changes in SARS-CoV-2 spread. Importantly, temporal variations of $C_\ell(t)$ are likely induced by non-pharmaceutical interventions (NPIs) [27], spontaneous behavioural changes and viral evolution [28] while the spatial covariance of $C_\ell(t)$ reflects the heterogeneity in the contact rate between departments/regions, in variant spread as well as in NPI implementation and compliance [29, 30]. The index $\tau$ denotes the time in days since the beginning of the infection and allows us to vary host infectivity over the duration of the infection. For this, $\zeta_\tau \in [0, 1]$ is the generation time probability mass (i.e. the relative contribution to infectivity) of the $\tau$-th day, for which we use the empirical serial interval distribution estimated by [31]. Finally, the $v_{\mathrm{T}} = 0.2$ ratio captures the reduction in viral transmission due to vaccination (mainly due to infection prevention [32]). The rationale of the Holling's type II functional response of the FOI (analogous to Michaelis-Menten kinetics) is elaborated in the Appendix of [21].

## Modelling vaccination

The French national vaccination campaign started in late December 2020. To incorporate the vaccine rollout into the model, we explicitly assumed that the vaccines partially prevent infection and critical COVID-19 by reducing the probability of being critically ill upon exposure, denoted by factor $v_{\mathrm{C}}$. We set vaccine coverage in each age class in the model using the official VAC-SI database [33].

Future vaccination rates were predicted using a linear regression for each age group trained on the previous 3 weeks of vaccination data. We assumed that vaccination begins with the older age classes and that all age classes have an arbitrary vaccine coverage threshold of 90%. If the coverage for an age class was ever over this threshold, the doses planned for this age class were redistributed to the next oldest age class.

To avoid the inflation of the number of parameters, we assumed that full vaccination only required a single dose. Another simplifying assumption is that the vaccine is instantaneously effective with an assumed permanent reduction in critical infection upon exposure of 90% ($v_{\mathrm{C}} = 0.1$), which is in the order of magnitude of the first real-life estimates available at the time of model development [32].

## Calculation

Parameter inference relies on a computed distance of the daily ICU admissions simulated by the model with respect to publicly reported data [34] after treatment for weekly seasonality. Let us denote the publicly produced figures for the number of ICU admissions in location $\ell$ (whether at the departmental, regional or national level) on calendar date $t$ by $a_{\ell,t}$. The weekly seasonality, being caused by systematic under-reporting on weekends and over-reporting during the beginning of the following work week, was smoothed out using 7-day rolling average followed by a Gaussian rounding (denoted by $\lfloor\cdot\rceil$). The observed data considered for the inference procedure is therefore $a_{\ell,t}^\circ := \left\lfloor\frac{1}{7}\sum_{\tau=0}^{6}a_{\ell,t-\tau}\right\rceil$ and let $\mathcal{T}$ be the set of calendar dates $t$ for which $a_{\ell,t}^\circ$ is computable.

Denoting by $a_{\ell,t}^{s}(\mathbf{x})$ the number of ICU admissions in location $\ell$ at time $t$ simulated by the model being parametrised with the set of parameter values $\mathbf{x}$, the log-likelihood of $\mathbf{x}$ given the observed data is computed as follows:

$$\log \mathcal{L}_{\ell,\mathcal{T}}(\mathbf{x}) := \log \mathbb{P}[(a_{\ell,t}^{\circ})_{t \in \mathcal{T}} | (a_{\ell,t}^{s}(\mathbf{x}))_{t \in \mathcal{T}}] := \sum_{t \in \mathcal{T}} a_{\ell,t}^{\circ} \log(a_{\ell,t}^{s}(\mathbf{x})) - \log(a_{\ell,t}^{\circ}!) - a_{\ell,t}^{s}(\mathbf{x}). \quad (2)$$

This definition holds if the distance between the model and the observation on a given date is seen as the random fluctuation of an integer-valued random variable. The random variable can reasonably be assumed to be Poisson-distributed. This assumption works well for small admission numbers because the population sizes of the investigated locations are large while the daily individual probability of being admitted in an ICU for COVID-19 is small. Assuming that the random fluctuations are independent across days and locations, the log-likelihood over a given set of dates $\mathcal{T}$ can be easily computed by summing over the daily log-likelihoods.

Parameters were then inferred under the Bayesian framework by running a Markov chain Monte Carlo (MCMC) algorithm—implemented in the `BayesianTools` R package [35]— over a set of 12,000 realisations of the model. The last 2,000 iterations were used to generate the median and the equal-tailed 95% credibility intervals for the parameters as well as the 95% forecasting range for the extrapolated time series of daily ICU admissions assuming a Poisson likelihood. The reproduction number $\mathcal{R}_0$ and the expectation and variance of the infection-to-hospitalisation delay were inferred at the nationwide level only, while all other parameters were independently fit for each sub-national administrative division. Details on the inferred parameters, their prior values and distributions are provided in Tables A, B, C, D and E in S1 Text and in [21].

We expected some of these parameters to change over time due a variety of factors including virus evolution (e.g. the increased transmissibility of the Alpha [28] and Delta [36] variants), public health interventions (e.g. lockdowns, curfews, limitations on businesses, etc.), social factors (e.g. school holidays), improvement in COVID-19 patient care, and variation in patient profiles. To account for this, we allowed for some parameters to be time-dependent by partitioning the time since the beginning of the epidemic and allowing each period to be associated with its own parameter set.

Parameter estimations were optimised based on the daily COVID-19-related critical care admissions from the COVID-19 hospital activity database (SI-VIC) [16]. In France, critically ill patients can be hospitalised either in intensive care units, continuous care units, or acute care units, with the three forming the critical care capacities [37]. For simplicity here, ICU refers to the wider category of critical care beds, as provided by SI-VIC. Furthermore, we assume the age distribution between localities to be fixed and based on the official demographics data [38].

## Communication

An automated cluster computing workflow refit the `COVIDici` model using daily updates of hospital, vaccination and testing data downloaded from the SI-VIC database, allowing a Shiny web application (see link in Introduction or [22] for source code) to communicate real-time results to the public. The original 2021 production version permitted users to visualise the combined past and future model fit by national, regional or departmental administrative unit for multiple epidemiological parameters, including ICU admissions, ICU occupancy, mortality (cumulative and daily), temporal reproductive number ($R_t$), infections (cumulative, daily and current), vaccination coverage and incidence for positive tests.

In 2022, a post-mortem version of the interface was deployed to allow for retrospective inspection of past forecasts with respect to a historical reference date. This version allows visualisation of all forecasts occurring prior to the reference date and includes basic evaluation metrics based on ICU overload (i.e. binarised ICU occupancy) and a colour-coded heatmap of hospital strain for varying forecast lengths and arbitrary saturation thresholds.

## Retrospective evaluation

Our assessment is based on original forecasts made by `COVIDici` between January 30, 2021 and December 2, 2021 (i.e. the first detected Omicron case in France), taken on a weekly basis to match with evaluation frameworks of the European and US Covid-19 Forecast Hubs. We focus here on ICU occupancy and only consider the regional and national levels while emphasising that many of the results are equally as valid on the departmental level, especially when they contain major urban areas. The following baseline models were evaluated using a rolling forecasting origin [39] starting on August 2, 2020:

- **ETS+ARIMA** is an ensemble of an ARIMA and an exponential smoothing (ETS) model fit using automated defaults in the `fable` package in R. It uses a log transformation of the rolling 7-day average of the ICU occupancy using only the data available to `COVIDici` to make its original forecast for that reference date.

- **AR-Lasso** is an auto-regressive (AR) machine learning type model that does not require stationarity. Lagged values from the previous 21 days are selected using the Least Absolute Shrinkage and Selection Operator (LASSO) [40] tuned using a 14-day time-series cross-validation as implemented in [41] to prevent data leakage and reduce overfitting. Prediction intervals were calculated using a bootstrap of the one-step ahead residuals from the training fit.

- **Naive** is a special case of an AR-1 implemented using the `fable` package. The point forecast is simply the last observed value and is optimal if the time series is a random walk [39].

## Standard metrics

We define "standard metrics" as those recommended by the US and European COVID-19 Forecast Hubs using the same 23 quantiles of the forecast distribution. Thus, point forecasts (based on the median) are evaluated with the absolute error (AE), individual prediction levels with the empirical coverage rate (ECR), and the forecast distributions with the weighted interval score (WIS). The WIS is a proper scoring rule that generalises the absolute error and gives penalties for interval spread as well as for over- and under-prediction. Formally speaking, for the true value of the target variable $y$, we have a forecast distribution $F$ with median $m$ that contains a set of $K$ prediction intervals whose respective upper ($u$) and lower ($l$) limits are the $\frac{\alpha}{2}$ and $1 - \frac{\alpha}{2}$ quantiles of the $F$. WIS is defined as the following weighted sum:

$$\text{WIS} = \frac{1}{K + 0.5} \left( 0.5 \cdot \mid y - m \mid + \sum_{k=1}^{K} \frac{\alpha_k}{2} \cdot \text{IS}_{\alpha_k}(F, y) \right) \quad (3)$$

where for a single interval $k$, the interval score is computed as

$$\text{IS}_{\alpha_k}(F, y) = (u - l) + \frac{2}{\alpha_k} \cdot (l - y) \cdot \mathbb{1}(y \leq l) + \frac{2}{\alpha_k} \cdot (y - u) \cdot \mathbb{1}(y \geq u) \quad (4)$$

with $\mathbb{1}$ being an indicator function and $\alpha_k$ the nominal coverage of interval $k$. Note that if only the median is included (i.e. $K = 0$) then the WIS simplifies to the AE. Furthermore, as the

number of equally spaced intervals increases, the WIS converges to the Continuous Ranked Probability Score. We refer readers to [42] for a deeper technical explanation.

All three standard metrics (AE, ECR, WIS) were calculated using the scoringutils package [43]. We used the package's default summary function (i.e. the mean) when aggregating only over geographic units. However, we use the median when aggregating over time which is not uncommon in the forecasting literature for COVID-19 [44–46] as it is more robust to the abnormally large errors that are common during the peaks of epidemiological waves. These exaggerated errors are clearly visible for COVIDici in Fig 2A and for ETS+ARIMA in Fig 2B

**A**

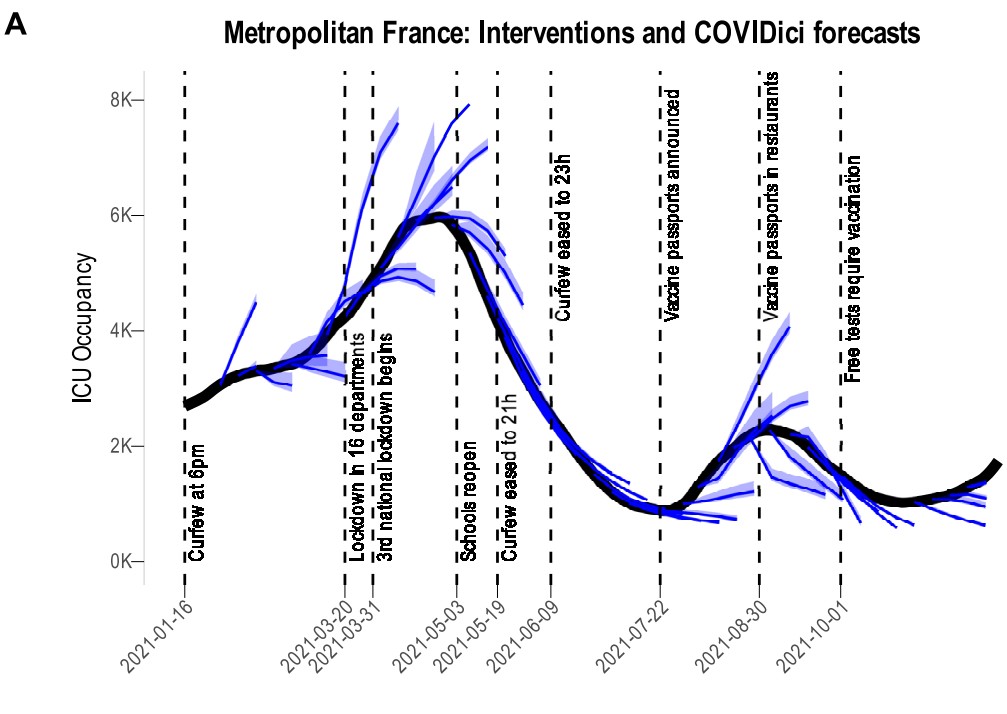

**B**

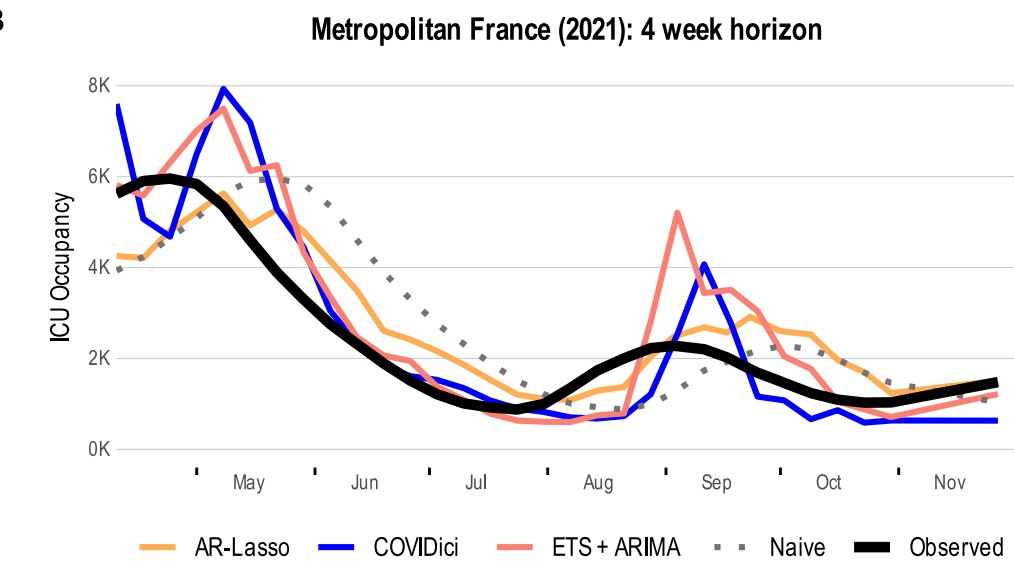

**Fig 2. Qualitative inspection of COVIDici forecasts.** A) Overlay of national forecasts of ICU occupancy produced by COVIDici with list of governmental interventions issued (national level only). B) All forecasters at the 4-week horizon plotted with the observed ICU occupancy for metropolitan France.

where its 4-week forecast horizon for metropolitan France more than doubled the observed ICU occupancy in early September 2021.

Summarising AE and WIS with the median have the drawback that it is more likely to reflect forecaster performance between waves rather than its ability to anticipate peaks, which is arguably more important depending out the forecaster's objective. Furthermore, AE and WIS tend to harshly punish large errors during wave peaks which can at least be partially explained by a survivor bias that occurs every time a public health policy is implemented (see Fig 2A for non-exhaustive list of national interventions in France), as well as spontaneous behavioural change. As artistically illustrated by Fig 3, this bias (i.e. the shaded area between the curves) corresponds to the difference between what would happen in absence of intervention (dashed curve) and what eventually is observed (the solid curve). While the magnitude of this bias at the peaks is counterfactual and subject to debate, several studies have indicated that even mild non-pharmaceutical interventions can have similar effects on curbing the spread of the virus compared to more severe ones [47, 48].

## Binary metrics

To more fairly evaluate performance during wave peaks, we consider ICU overload (i.e. binarised ICU occupancy), which we expect to be more robust against over-predictions. This requires introducing arbitrary capacity thresholds which we define as the percentage of the ICU occupancy observed in the geographical unit in the first wave in 2020. For point forecasts, we consider the proportion of incorrect forecasts of an outcome given that outcome was observed. Following the convention that lower scores are better, we define:

$$1 - \text{Sensitivity} \quad = \quad \frac{\text{\# of incorrect forecasts of overload}}{\text{\# of observed overloads}} \tag{5}$$

$$1 - \text{Specificity} \quad = \quad \frac{\text{\# of incorrect forecasts of underload}}{\text{\# of observed underloads}} \tag{6}$$

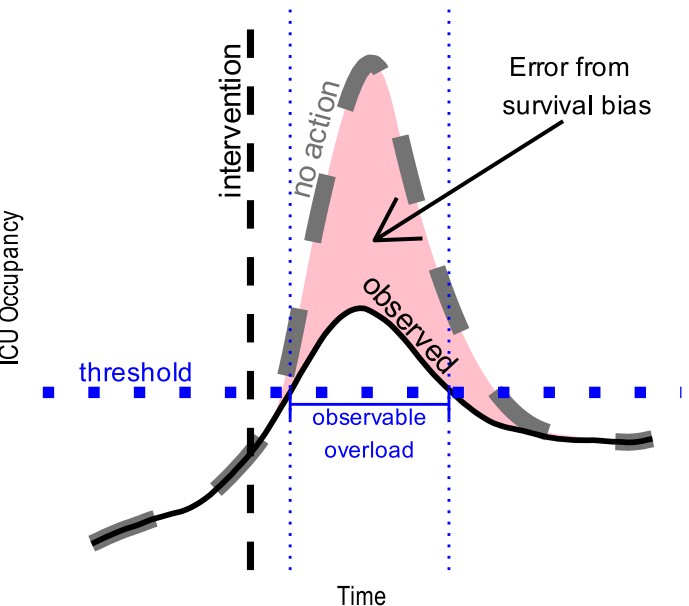

**Fig 3. Artistic representation of the survivor bias that may occur after a restrictive governmental intervention.**
The dashed curve is the counterfactual ICU occupancy that would have occurred if the intervention did not happen. The solid curve is the ICU occupancy that we observed because the intervention did occur.

For forecast distributions of ICU overload, we use the Brier score, which is the mean squared error of the binary overload outcome (i.e. 0 or 1) and the mass of the prediction interval above the arbitrary threshold. Formally, this is defined as:

$$\text{Brier score} = \frac{1}{N}\sum_{n=1}^{N}(f_n - y_n)^2 \tag{7}$$

where $f_n$ is the predicted probability of overload event $y_n \in \{0, 1\}$ with $n = 1, \ldots, N$ denoting all the events in the scope of the evaluation. The predicted probability $f_n$ can be approximately calculated as

$$f_n \approx 1 - \min\{q \mid q \geq \text{threshold}\} \tag{8}$$

where $q$ is the collection of quantiles of the forecast distribution and the threshold is fixed and arbitrary. We present binary metrics separately for periods of observed overload and underload because anticipation of overload is widely considered more important in a hospitalisation surveillance system.

## Statistical comparisons of forecast metrics

Many statistical approaches for COVID-19 forecast comparison have been proposed that focus primarily on producing p-values for hypothesis testing. Some notable examples include Diebold-Mariano (DM) tests [49], permutation tests (implemented in scoringutils for mean only), Wilcoxon signed-rank test (also implemented in scoringutils) and Mood's Median test [44]. While all these approaches can be quite suitable for many forecasting situations, the latter is not applicable to comparing groups paired by target date and the former three may be negatively affected when comparing two forecast error distributions with different shapes [50–52]. Recall that abnormally large errors in COVIDici and ETS+ARIMA at the wave peaks indeed create much heavier tails in their respective forecast error distributions compared to the Naive and AR-Lasso models (for example see Fig B in S1 Text). Furthermore, all these tests ignore potential serial dependencies when aggregating across geographic units while only the DM test accounts for temporal auto-correlation.

The goal of the statistical evaluation presented here is to retrospectively identify significant patterns in the forecast performance of COVIDici, and, as such, is exploratory in nature not confirmatory [53, 54]. Given the multitude of potential comparisons to be made across varying geographical units, forecast horizons and time, we present statistical comparisons between two models by focusing primarily on confidence intervals (CIs) that can be integrated into interpretable graphics. In particular, we implement CIs using a non-parametric bootstrap with 10, 000 replicates for the ratio of the aforementioned summary metrics. If the summary metric aggregates over only time then forecast dates are independently resampled with replacement and if aggregation occurs over both space and time then both geographic units and forecast dates are simultaneously resampled. By consequence, potential serial dependences across space and time are ignored.

To account for skewness in the distribution of the bootstrapped test statistic we apply bias-corrected and accelerated (BCa) CIs [55]. The main advantage of the BCa bootstrap approach is that it is general enough to be applied to every metric regardless of potential skewness in one or both of the forecast error distributions. This adjustment also ensures that the CIs are second-order accurate and transformation invariant, which is a convenient property when presenting plots on the log scale. The main disadvantage is that we resample independently in a manner that ignores potential spatial and temporal dependency. The acceleration parameter that accounts for skewness is calculated using a finite-sample jackknife. A formal explanation

of the confidence interval implementation is notationally tedious so it is only provided in S1 Text. However, convenient practical implementation is done using the nptest package [56] to produce CIs at the 95% level.

P-values are calculated post hoc by CI inversion [57, 58], which simply means that the p-value for an arbitrary value (i.e. 1 for a ratio comparison and 0 for a log-ratio comparison) is defined as the smallest $\alpha$ such that the corresponding $1 - \alpha$ CI does not contain that value. Being an exploratory data analysis, we designate important levels for p-values at $\alpha = 0.001$, 0.01, 0.05 and 0.1, but emphasise to the reader that these are intended to be interpreted in conjunction with their associated $1 - \alpha$ CIs rather than being formal hypothesis tests where it is preferable to resample under the assumption that the null hypothesis is true. Practically speaking, the null hypothesis is that the ratio of summary metric statistics is 1 (0 on the log scale) and is rejected when this value not is contained in the CI.

## Results

To qualitatively inspect the forecasts produced by COVIDici relative to the observed (smoothed) ICU occupancy time series, we first refer to Fig 2A. It is noteworthy that COVIDici performs better on the trailing side of each wave than the leading side, a phenomenon that is also apparent on the regional and departmental level (see Post-mortem version of the Shiny app linked in the introduction). It is not completely clear why this is case but this possibly indicates more uncertainty in changes in social behaviour and effectiveness of governmental interventions as the number of infections surges. Furthermore, the turning points at the wave peaks and wave troughs (i.e. local maxima/minima) are characterized by large misses. To a certain extent this is to be expected since COVIDici is informed by changes in ICU admissions which may take approximately 2 weeks to manifest following a surge or decline in new infections.

Fig 2B visualises the national forecasts at the 4-week horizon for COVIDici and the 3 baseline models. The ensemble of ARIMA+ETS exhibits a similar behaviour to COVIDici where it greatly over-predicts at the top of the waves and shows the same delay in detecting the beginning of a new one. AR-Lasso and the Naive model on the other hand appear to predict a right-shifted version of the observed ICU occupancy time series. For the Naive model this shift is exact and equal to the length of the forecast horizon.

Standard metrics for ICU occupancy are contained in Fig 4. Fig 4A shows the overall empirical coverage rates across all geographic units, forecast dates and forecast horizons where the dashed line is optimal. Only ETS+ARIMA appears to exhibit a reasonably calibrated forecast distribution, followed by AR-Lasso and the Naive model. The worst calibration by far are the prediction intervals for COVIDici which are almost horizontal in Fig 4A, indicating that the prediction intervals are far too narrow. This occurred because the forecast distribution implemented by COVIDici is merely the credible interval for a model parameter, i.e. ICU occupancy, rather than a proper prediction interval describing the uncertainty impacting a future observation. As a result, COVIDici's performance relative to all metrics for forecast distributions will be negatively affected as can be visualised in Fig 4B which shows the mean WIS (scaled by the naive model) across all geographic units over time. Large misses near wave peaks are evident for COVIDici and ETS+ARIMA but not AR-Lasso, which appears to be more consistent.

Fig 5 shows a forest plot by region of the median AE of COVIDici relative to the baseline models. At the two-week horizon COVIDici outperformed the Naive baseline in metropolitan France ($p = .06$), Bretagne ($p = .04$), Hauts-de-France ($p = .01$) and Île-de-France ($p = .08$), but actually underperformed the ensemble ETS+ARIMA baseline in metropolitan France

**A**  **Coverage rates by region**

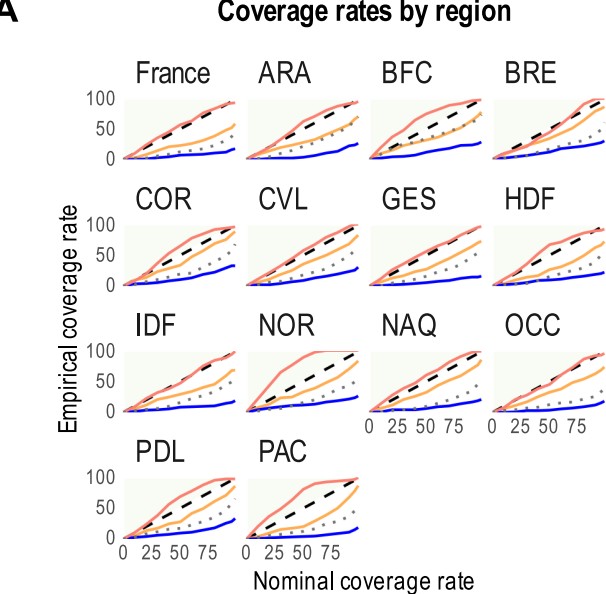

**B**  **Mean WIS across all geographic units over time**

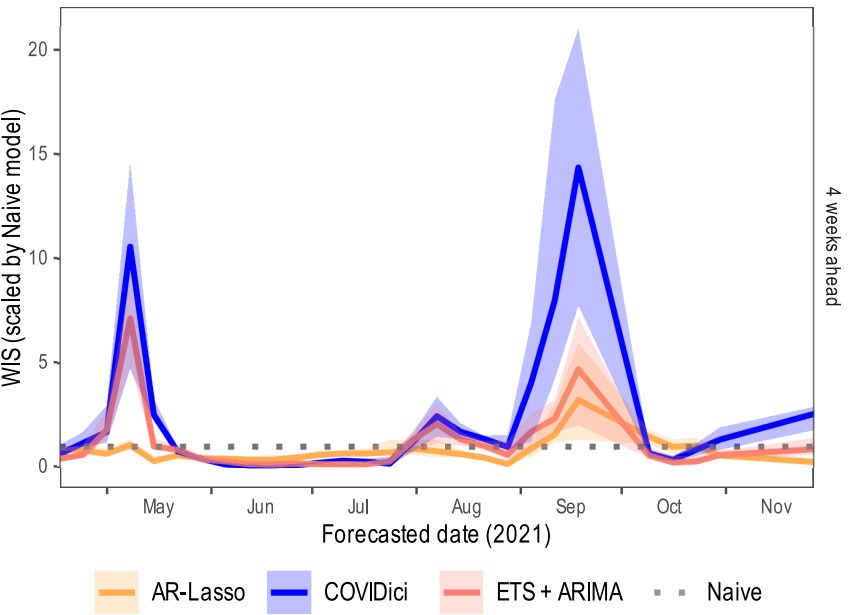

**Fig 4. Standard evaluation metrics for ICU occupancy.** A) Empirical coverage rate across all geographic units and forecast horizons (dashed line is optimal). Auvergne-Rhône-Alpes = ARA, Bourgogne-Franche-Comté = BFC, Bretagne = BRE, Centre-Val de Loire = CVL, Corse = COR, Grand Est = GES, Hauts-de-France = HDF, Île-de-France = IDF, Normandie = NOR, Nouvelle-Aquitaine = NAQ, Occitanie = OCC, Pays de la Loire = PDL, Provence-Alpes-Côte d'Azur = PAC. B) Mean weighted interval score (WIS) of all geographic units over time for the four-week forecast horizon scaled by the Naive model. Shaded areas represent 95% bias-corrected and accelerated (BCa) bootstrap confidence intervals with 10000 replicates per target date.

## Median absolute error (AE) of COVIDici relative to baselines

| Region | n | ratio | [95% CI] | p value | | n | ratio | [95% CI] | p value | |
|---|---|---|---|---|---|---|---|---|---|---|
| | | | | **2 weeks ahead** | | | | | **4 weeks ahead** | |
| France | 38 | 0.49 | [0.25, 0.97] | .04 | * | 31 | 0.59 | [0.3, 1.02] | .06 | • |
| | | 1.83 | [0.96, 3.12] | .06 | • | | 1.07 | [0.53, 1.78] | .90 | |
| | | 0.97 | [0.5, 2.22] | .95 | | | 0.86 | [0.44, 1.72] | .73 | |
| CVL | 38 | 1.21 | [0.67, 2.17] | .40 | | 31 | 0.51 | [0.25, 0.98] | .04 | * |
| | | 1.65 | [0.72, 3.06] | .24 | | | 0.79 | [0.52, 1.7] | .59 | |
| | | 1.55 | [0.79, 3.26] | .16 | | | 0.46 | [0.25, 1] | .05 | • |
| ARA | 38 | 0.71 | [0.47, 1.22] | .19 | | 31 | 0.57 | [0.27, 1.12] | .17 | |
| | | 1.19 | [0.84, 2.05] | .48 | | | 0.62 | [0.29, 0.82] | .008 | ** |
| | | 1.07 | [0.76, 1.96] | .56 | | | 0.47 | [0.32, 0.94] | .03 | * |
| NOR | 38 | 0.73 | [0.43, 1.27] | .23 | | 31 | 0.58 | [0.28, 0.87] | .02 | * |
| | | 0.82 | [0.55, 1.32] | .26 | | | 0.74 | [0.39, 1.33] | .22 | |
| | | 0.87 | [0.54, 1.34] | .33 | | | 0.61 | [0.37, 0.94] | .05 | * |
| BRE | 38 | 0.64 | [0.43, 0.98] | .04 | * | 29 | 0.59 | [0.3, 1.04] | .05 | • |
| | | 0.88 | [0.56, 1.4] | .63 | | | 0.66 | [0.39, 1.11] | .12 | |
| | | 1.04 | [0.69, 1.57] | .93 | | | 0.64 | [0.4, 1.14] | .14 | |
| OCC | 38 | 0.65 | [0.35, 1.44] | .36 | | 31 | 0.61 | [0.3, 0.94] | .04 | * |
| | | 1.16 | [0.72, 2.32] | .49 | | | 0.63 | [0.33, 1.16] | .13 | |
| | | 1.52 | [0.84, 3.5] | .19 | | | 0.89 | [0.44, 1.45] | .56 | |
| HDF | 38 | 0.57 | [0.3, 0.89] | .01 | * | 30 | 0.62 | [0.34, 1.12] | .12 | |
| | | 1 | [0.62, 1.64] | .92 | | | 0.76 | [0.48, 1.21] | .26 | |
| | | 1.15 | [0.76, 1.58] | .44 | | | 0.51 | [0.35, 0.93] | .04 | * |
| PAC | 38 | 1.03 | [0.48, 2.15] | .83 | | 31 | 0.65 | [0.32, 2.15] | .61 | |
| | | 2.41 | [1.32, 5.39] | .009 | ** | | 0.84 | [0.43, 2.54] | .69 | |
| | | 1.4 | [0.79, 2.51] | .20 | | | 0.75 | [0.41, 2.05] | .64 | |
| NAQ | 38 | 1 | [0.58, 1.83] | .86 | | 31 | 0.79 | [0.39, 1.38] | .46 | |
| | | 1.35 | [0.88, 2.05] | .23 | | | 1.08 | [0.56, 1.64] | .83 | |
| | | 1.44 | [0.78, 2.67] | .20 | | | 0.88 | [0.41, 1.48] | .59 | |
| IDF | 38 | 0.73 | [0.38, 1.05] | .08 | • | 31 | 0.81 | [0.37, 1.51] | .58 | |
| | | 1.79 | [1.11, 3.09] | .02 | * | | 1.05 | [0.61, 1.97] | .73 | |
| | | 1.35 | [0.85, 2.31] | .15 | | | 0.84 | [0.47, 1.49] | .59 | |
| PDL | 38 | 0.76 | [0.53, 1.14] | .30 | | 31 | 0.83 | [0.35, 1.33] | .32 | |
| | | 1.25 | [0.75, 1.96] | .37 | | | 1.1 | [0.52, 1.86] | .78 | |
| | | 0.92 | [0.63, 1.41] | .69 | | | 0.8 | [0.37, 1.14] | .15 | |
| BFC | 38 | 1.15 | [0.66, 2.86] | .44 | | 31 | 0.86 | [0.41, 1.4] | .62 | |
| | | 1.47 | [0.99, 3.08] | .07 | • | | 0.8 | [0.41, 1.52] | .42 | |
| | | 1.24 | [0.85, 1.94] | .23 | | | 0.63 | [0.34, 1.1] | .12 | |
| GES | 38 | 0.94 | [0.53, 1.59] | .84 | | 31 | 1.15 | [0.56, 2.07] | .47 | |
| | | 1.38 | [0.97, 2.17] | .07 | • | | 1.14 | [0.77, 1.84] | .27 | |
| | | 1.33 | [0.85, 1.94] | .19 | | | 0.81 | [0.47, 1.33] | .35 | |
| COR | 38 | 1.34 | [0.85, 2.63] | .21 | | 30 | 1.29 | [0.83, 2.45] | .24 | |
| | | 1.56 | [0.9, 2.32] | .11 | | | 1.28 | [0.75, 1.8] | .38 | |
| | | 1.37 | [0.8, 3.26] | .24 | | | 1.1 | [0.54, 2.05] | .89 | |

Signif. codes :
*** p < .001
** .001 ≤ p < .01
* .01 ≤ p < .05
• .05 ≤ p < .1

Favors model ← → Favors baseline

◆ Naive ◆ ETS + ARIMA ◆ AR-Lasso

**Fig 5. Median absolute error (AE) for `COVIDici` relative to other models.** Ratio = median AE for `COVIDici` / median AE for baseline model. 95% confidence interval (CI) is the bias-corrected and accelerated (BCa) bootstrap confidence interval generated by a nonparametric bootstrap with 10000 replicates. The p value is the smallest alpha such that 1 is not contained in the corresponding $1 - \alpha$ CI. See Fig 4 for region code definitions.

($p = .06$), Provence-Alpes-Côte d'Azur ($p = .009$), Île-de-France ($p = .02$), Bourgogne-Franche-Comté ($p = .07$) and Grand Est ($p = .07$). However, at the four-week horizon COVIDici was never statistically outperformed by any baseline model yet did better than at least one of the baselines in 7 of the 14 considered geographic units: compared the Naive model in metropolitan France ($p = .06$), Centre-Val de Loire ($p = .04$), Normandie ($p = .02$), Bretagne ($p = .05$) and Occitanie ($p = .04$); compared to ETS+ARIMA in Auvergne-Rhône-Alpes ($p = .008$); and compared to AR-Lasso in Centre-Val de Loire ($p = .05$), Auvergne-Rhône-Alpes ($p = .03$), Normandie ($p = .05$) and Hauts-de-France ($p = .04$). This reflects a modest but consistent improvement in forecast performance at the longer forecast horizon. Furthermore, it is unlikely that these results at the four-week horizon occurred simply due to a multiple testing problem since if the null hypothesis is true then the p-values should be uniformly distributed and all significant tests favored COVIDici.

Fig 6 shows a similar forest plot except using median WIS as the summary metric. The overly-narrow prediction intervals clearly degraded COVIDici's performance here as it never outcompeted any of the baselines at the two-week horizon and only performed better compared to ETS+ARIMA in Normandie ($p < .001$). It was also outperformed by at least one of the baselines in 10 of 14 considered geographic units.

Binarised metrics for ICU overload are shown in Fig 7. The left column shows the raw summary metrics without CIs while the right column shows the performances of the baselines relative to COVIDici including their respective bootstrapped CIs. The relative metrics are on the log scale for more visual clarity and are interpreted to be statistically significant for a given capacity threshold when the CI of the baseline model does not contain zero (i.e. the horizontal blue line).

COVIDici performed poorly at the two-week horizon for nearly all metrics presented here except relative to the Naive model when overload was observed at capacity thresholds greater than approximately 0.75. On the other hand, on the four-week horizon, COVIDici outperforms AR-Lasso and the Naive model for most thresholds considered when overload was observed. For the % of incorrect forecasts at the longer horizon we fail to detect any statistical difference between COVIDici and ETS+ARIMA for most capacity thresholds. For Brier score, COVIDici's performance was somewhat degraded by its overly-narrow prediction intervals just as was the case for other aforementioned distribution metrics.

To summarise, COVIDici tended to exhibit improved forecasting performance at the four-week horizon than the two-week horizon relative to the considered baseline models. However, its prediction intervals were far too narrow, which led to poorer performance relative to metrics for forecast distributions. In terms of median AE it either performed comparably or better than all baselines at the four-week horizon for all considered sub-national geographic units. In terms of correctly predicting ICU overload, COVIDici was comparable to the ETS+ARIMA ensemble model at the four-week horizon and consistently outperformed the Naive and AR-Lasso baselines for most of the thresholds considered.

## Discussion

### Modelling and forecasting

Mathematical modelling was vital to understanding the COVID-19 epidemic in real-time as it unfolded. In France, it led to the anticipation of hospital dynamics in the first epidemic wave [59] and the emergence of the Delta variant [36]. However, most of these studies were performed at a national level and/or at a single time point, which made their impact limited from a public health point of view. To our knowledge, there were only two continuously running forecasting models in France in 2021 at the subnational level: COVIDici and an ensemble of

## Median weighted interval score (WIS) of COVIDici relative to baselines

| Region | n | ratio | [95% CI] | p value | | | n | ratio | [95% CI] | p value | |
|---|---|---|---|---|---|---|---|---|---|---|---|---|
| | | | **2 weeks ahead** | | | | | | **4 weeks ahead** | | |
| France | 38 | 0.66 | [0.33, 1.53] | .33 | | | 31 | 0.7 | [0.31, 1.29] | .28 | |
| | | 2.23 | [1.34, 3.95] | .007 | ** | | | 1.43 | [0.82, 2.18] | .17 | |
| | | 1.36 | [0.65, 3.56] | .44 | | | | 1.01 | [0.45, 2.3] | > .99 | |
| CVL | 38 | 1.59 | [0.74, 2.95] | .19 | | | 31 | 0.61 | [0.23, 1.54] | .28 | |
| | | 1.33 | [0.59, 2.19] | .45 | | | | 1.03 | [0.58, 1.94] | .89 | |
| | | 2.07 | [0.98, 3.79] | .05 | • | | | 0.58 | [0.28, 1.39] | .23 | |
| OCC | 38 | 0.72 | [0.38, 1.94] | .48 | | | 31 | 0.65 | [0.33, 1.19] | .21 | |
| | | 1.38 | [0.9, 3.03] | .16 | | | | 1.1 | [0.61, 1.89] | .76 | |
| | | 2.27 | [1.22, 5.69] | .01 | * | | | 1.31 | [0.64, 2.44] | .34 | |
| NOR | 38 | 1.04 | [0.52, 2.13] | .97 | | | 31 | 0.74 | [0.29, 1.24] | .25 | |
| | | 0.9 | [0.58, 1.32] | .46 | | | | 0.5 | [0.28, 0.76] | <.001 | *** |
| | | 1.51 | [0.9, 2.43] | .11 | | | | 0.96 | [0.57, 1.62] | .75 | |
| PAC | 38 | 1.36 | [0.55, 3.42] | .39 | | | 31 | 0.78 | [0.37, 2.77] | .79 | |
| | | 1.37 | [0.75, 2.27] | .30 | | | | 0.88 | [0.53, 2.2] | .78 | |
| | | 2.1 | [1.11, 4.19] | .02 | * | | | 1.19 | [0.61, 3.39] | .74 | |
| BRE | 38 | 0.76 | [0.38, 1.21] | .18 | | | 29 | 0.81 | [0.41, 1.64] | .60 | |
| | | 1.14 | [0.59, 1.66] | .64 | | | | 0.79 | [0.42, 1.19] | .22 | |
| | | 1.38 | [0.72, 2.05] | .31 | | | | 0.95 | [0.56, 1.94] | .88 | |
| ARA | 38 | 0.98 | [0.53, 1.6] | .72 | | | 31 | 0.87 | [0.35, 1.89] | .66 | |
| | | 1.2 | [0.81, 1.85] | .45 | | | | 0.89 | [0.43, 1.22] | .27 | |
| | | 1.4 | [0.88, 2.32] | .17 | | | | 0.68 | [0.42, 1.42] | .40 | |
| HDF | 38 | 0.75 | [0.34, 1.37] | .30 | | | 30 | 0.91 | [0.39, 1.66] | .78 | |
| | | 1.24 | [0.74, 1.93] | .41 | | | | 1.09 | [0.58, 1.64] | .65 | |
| | | 1.82 | [1.15, 2.4] | .02 | * | | | 0.83 | [0.44, 1.48] | .48 | |
| NAQ | 38 | 1.53 | [0.77, 2.87] | .15 | | | 31 | 0.97 | [0.47, 1.81] | .87 | |
| | | 1.7 | [1.08, 2.56] | .02 | * | | | 1.46 | [0.86, 2.54] | .13 | |
| | | 2.48 | [1.25, 4.34] | .02 | * | | | 1.36 | [0.65, 2.72] | .29 | |
| IDF | 38 | 1.02 | [0.45, 1.66] | .73 | | | 31 | 1.12 | [0.43, 2.68] | .64 | |
| | | 1.95 | [1.31, 3.1] | .006 | ** | | | 1.48 | [0.96, 2.67] | .06 | • |
| | | 1.95 | [1.19, 2.96] | .01 | * | | | 1.07 | [0.57, 2.18] | .84 | |
| PDL | 38 | 0.92 | [0.56, 1.62] | .82 | | | 31 | 1.18 | [0.42, 2.14] | .89 | |
| | | 1.02 | [0.64, 1.65] | .94 | | | | 1.03 | [0.45, 1.86] | .89 | |
| | | 1.24 | [0.79, 2.1] | .45 | | | | 1.15 | [0.53, 2.07] | .51 | |
| BFC | 38 | 1.6 | [0.77, 3.4] | .20 | | | 31 | 1.27 | [0.44, 2.12] | .52 | |
| | | 1.3 | [0.78, 2.18] | .23 | | | | 1.09 | [0.53, 1.67] | .77 | |
| | | 1.65 | [1.17, 2.55] | .01 | * | | | 0.82 | [0.36, 1.72] | .64 | |
| GES | 38 | 1.34 | [0.63, 2.42] | .46 | | | 31 | 1.75 | [0.62, 3.31] | .27 | |
| | | 1.41 | [0.89, 2.3] | .16 | | | | 1.57 | [0.82, 2.23] | .16 | |
| | | 2.15 | [1.37, 3.23] | .007 | ** | | | 1.08 | [0.49, 1.73] | .85 | |
| COR | 38 | 2.24 | [1.47, 4.71] | .001 | ** | | 30 | 1.81 | [1.09, 4.15] | .03 | * |
| | | 1.73 | [1.13, 3.13] | .01 | * | | | 1.71 | [1.08, 2.51] | .02 | * |
| | | 2.5 | [1.47, 5.52] | <.001 | *** | | | 1.76 | [0.96, 3.33] | .07 | • |

Signif. codes :
*** p < .001
** .001 ≤ p < .01
* .01 ≤ p < .05
• .05 ≤ p < .1

Favors model Favors baseline

◆ Naive ◆ ETS + ARIMA ◆ AR-Lasso

**Fig 6. Median weighted interval score of `COVIDici` relative to other models.** Ratio is the median weighted interval score (WIS) for `COVIDici` divided by median WIS using all other models as baselines. Each 95% confidence interval (CI) is based on the bias-corrected and accelerated (BCa) bootstrap confidence interval generated by nonparametric bootstrap with 10000 replicates. The p value is the smallest alpha such that 1 is not contained in the corresponding $1 - \alpha$ CI. See Fig 4 for region code definitions.

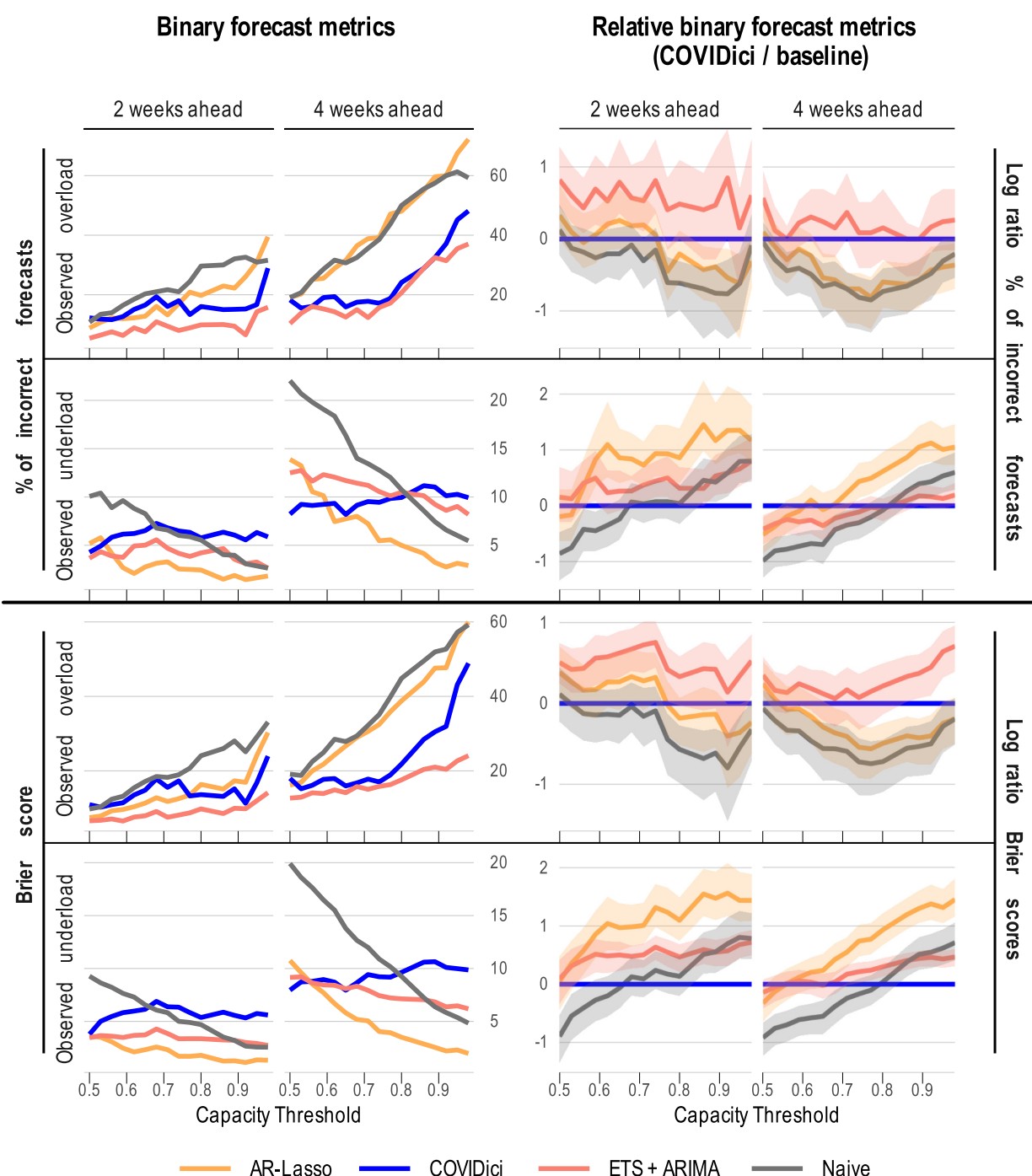

**Fig 7. Binary metrics for ICU occupancy overload by capacity threshold.** Metrics are conditioned separately for dates where ICU overload was observed or not observed respectively relative to arbitrary capacity thresholds. Thresholds are defined as a proportion of the peak ICU occupancy observed in a geographic unit. Shaded areas correspond to 95% bias-corrected and accelerated (BCa) confidence intervals based on 10000 replicates of a non-parametric bootstrap that assumes both spatial and temporal independence. Shaded areas completely below the blue line indicate that `COVIDici` statistically outperformed that model for that metric, threshold and forecast horizon.

statistical models implemented by the Pasteur Institute [10]. These two models were built on two types of approaches each with different strengths and limitations. For the latter this meant reasonable forecasting accuracy but only for relatively short forecast horizons.

In addition to real-time sub-national forecasting, a key feature of `COVIDici` was to offer visualisation of numerous unobservable indicators that can only be inferred through an underlying mechanistic model (e.g. the estimate of all active infections, reproductive number, etc.). This is an asset from a popularisation point of view, but it raises issues because estimates can be strongly biased in areas with low population density.

Furthermore, it is important to distinguish between two types of forecasts. Some, like `COVIDici`, attempt to project mechanistically what would happen if transmission remains identical, i.e. assuming transmission dynamics equal as those observed in the past weeks. Others, especially the ones built on machine learning, try to extrapolate phenomenologically the pattern of the time series by being sensitive to the most recent changes. In terms of guiding public health decision-making, both are complementary since the former is robust in the long term and can incorporate expected events (such as planned interventions or viral variant replacement), while the latter is accurate on the short term.

## Limitations

It is worthwhile to point out that while this evaluation of `COVIDici` is retrospective, its development, utilisation and updating occurred entirely during the pandemic. A tremendous number of models were proposed during that time that recommended forecasting various COVID-19 indicators using different approaches, and auxiliary data. However, all such proposals (`COVIDici` included) lacked the benefit of hindsight and were constrained by implementation feasibility and uncertainty over how well they would generalise into future stages of the pandemic. Thus, we emphasise that the following four limitations identified by our evaluation should be interpreted from this practical perspective.

1. The prediction intervals were far too narrow which obfuscated any performative advantages that it had when considering distributional evaluation metrics (e.g. WIS and Brier score). Future attention must be given to creating a better calibrated prediction interval under acceptable computational expense.

2. Fitting `COVIDici` both nationally and sub-nationally, for 13 regions and 101 departments, using MCMC is very computationally expensive. At the time the `COVIDici` was ended, the model fitting update on a high-performance cluster using 115 cores took over 4 hours to complete. This meant that refitting the model was rarely feasible from a practical perspective, which is problematic when trying to incorporate additional model fits for different scenarios or it is discovered that a server had been down for maintenance or that a programming oversight led to only saving 3 forecast quantiles.

3. `COVIDici` exhibited relatively weak performance up to the two-week horizon to predict ICU occupancy compared to statistical modelling approaches. However, this was expected as `COVIDici` only uses hospital data to update its inference and there is a nearly two-week delay between infection and hospital admission [60]. This decreased performance at the one- and two-week horizons relative to a benchmark is also consistent with findings regarding other mechanistic compartmental models predicting cumulative deaths at the national level in the United States [61].

4. From a pure forecasting perspective, `COVIDici` only has modest improvements at the four-week horizon compared to rather basic baseline models such as the Naive model. We

emphasise though that this is not the whole story because, contrary to statistical models, it provides a full epidemiological insight in the form of unobservable estimated parameters such as spatialised current prevalence and attack rates.

Regarding limitations to our retrospective evaluation, the results found here may not be immediately generalizable everywhere. The implementation of mechanistic compartmental models requires many simplifying biological assumptions that depend on expert opinion which exists in the context of what is known about the disease at that time. If `COVIDici` were to be built again today one would likely not use all the same assumptions, e.g. full immunity after recovery from infection. Furthermore, the effects of viral evolution, governmental interventions and spontaneous social changes are very dependent on the time period during which the evaluation is framed. As a result it is hard to comment on the reproduceablilty of these results for other locations at different times.

## Conclusion

Many countries are decreasing their investment in epidemic surveillance, and some rely on statistical model forecasting with the inclusion of new predictors, such as that from wastewater data. However, there is still room for compartmental model forecasts like `COVIDici` that can rely on variables with a high level of sampling such as hospital admissions data.

Regarding SARS-CoV-2, future extensions of `COVIDici` would require updating the model to account more precisely for the diversity in immune protection among individuals, given the number of natural infections since the evolution of the Omicron variants [17]. However, existing non-Markovian models suggest that this is feasible [60]. In several ways `COVIDici`'s forecasting potential was under-exploited. It already incorporated variations in vaccine coverage but, thanks to its mechanistic nature, it could also readily include planned events such as school holidays or early predictors of variations in reproduction numbers such as viral evolution or weather.

The main strength of `COVIDici` was improved accuracy at the four-week horizon for point forecasts of ICU occupancy compared to the statistical baseline models, especially during the trailing edge of waves. For anticipating wave peaks, `COVIDici` had one of the best overall performances with respect to the trade-off between the correct prediction rate of observed overload and observed underload on the four-week horizon. The baseline model based on machine learning (i.e. AR-Lasso) on the other hand failed to reasonably anticipate overload, despite relatively optimistic performance in terms of more standard metrics for continuous variables such as WIS. This should serve as a cautionary example for similar models that avoid large errors by avoiding large predictions at peaks of the waves. Systematically avoiding pessimistic predictions may improve the resulting evaluation score, but it also may complicate decision-making regarding non-pharmaceutical government interventions which is a common goal when forecasting hospital strain. To detect models exhibiting such undesirable behaviour, it seems reasonable to consider alternative evaluation metrics based on ICU overload (i.e. binarised ICU occupancy) especially when the evaluation period contains frequent interventions in the training sets.

Those building surveillance systems in future pandemics may consider the application of non-Markovian compartmental models despite their limitations. This is evidenced by the fact that it is clearly feasible since `COVIDici` was successfully developed and deployed in real-time during the COVID-19 pandemic and that long-term forecasting utility, albeit modest, has subsequently been established at the national and sub-national levels for ICU occupancy.

## Supporting information

**S1 Text. Technical details for real-time forecasting of COVID-19 in France using a non-Markovian mechanistic model.**
(PDF)

## Acknowledgments

We want to thank Santé publique France for providing timely open-source data, the South Green computational platform/IRD for access to their cluster and the French Institute of Bioinformatics for support hosting the `COVIDici` web application.

## Author Contributions

**Conceptualization:** Samuel Alizon, Mircea T. Sofonea.

**Data curation:** Alexander Massey, Corentin Boennec.

**Formal analysis:** Alexander Massey.

**Funding acquisition:** Samuel Alizon, Mircea T. Sofonea.

**Investigation:** Mircea T. Sofonea.

**Methodology:** Mircea T. Sofonea.

**Project administration:** Christophe Blanchet, Samuel Alizon, Mircea T. Sofonea.

**Software:** Alexander Massey, Corentin Boennec, Claudia Ximena Restrepo-Ortiz, Christophe Blanchet, Mircea T. Sofonea.

**Supervision:** Mircea T. Sofonea.

**Validation:** Alexander Massey.

**Visualization:** Alexander Massey, Corentin Boennec.

**Writing – original draft:** Alexander Massey, Corentin Boennec, Samuel Alizon, Mircea T. Sofonea.

**Writing – review & editing:** Alexander Massey, Samuel Alizon, Mircea T. Sofonea.

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
