## [Decision Letter · Decision Letter 0]

17 Oct 2023

Dear Dr. Massey,

Thank you very much for submitting your manuscript "Real-time forecasting of COVID-19-related hospital strain in France using a non-Markovian mechanistic model" for consideration at PLOS Computational Biology.

As with all papers reviewed by the journal, your manuscript was reviewed by members of the editorial board and by several independent reviewers. In light of the reviews (below this email), we would like to invite the resubmission of a significantly-revised version that takes into account the reviewers' comments.

We cannot make any decision about publication until we have seen the revised manuscript and your response to the reviewers' comments. Your revised manuscript is also likely to be sent to reviewers for further evaluation.

Sincerely,

Eric Lofgren, MSPH, PhD

Academic Editor

PLOS Computational Biology

Virginia Pitzer

Section Editor

PLOS Computational Biology

Reviewer's Responses to Questions

**Comments to the Authors:**

Reviewer #1: This article conducts a comparative study on forecasting critical care unit overload pertaining to COVID-19 in sub-national groupings in France. A non-Markovian compartmental model is presented and compared against standard forecasting models of Auto Regression, exponential smoothing, and Autoregressive Integrated Moving Average (ARIMA). The article is written well, the data and code underlying the article are included in an accessible repository, and many interesting insights are presented. However, there is a lack of quantitative results supporting the articles conclusions. The primary weakness of the article in its current form is that while this is a comparative study, tests have not been applied, or at least properly reported on, to support the presence or lack of presence of statistically significant differences between the three forecasting methods. Therefore, I recommend this article for a Major Revision pending consideration of the following comments.

**Primary Comments**

1) Tests for significance should be applied to determine if any differences between the Non-Markovian compartmental model and the three tested methods are statistically significant. The null hypotheses driving the comparisons should also be explicitly stated.

* Additionally, the significance level utilized in the study should be presented in the “Materials and methods” section

2) The study is not properly situated within the literature.

2.1. Similar to the supporting arguments that forecasting differences occur at local versus national interventions, there is also evidence that forecasts have statistically significant differences at varied geographic scale for COVID-19. Some suggestions follow:

Barría-Sandoval, C., Ferreira, G., Benz-Parra, K., & López-Flores, P. (2021). Prediction of confirmed cases of and deaths caused by COVID-19 in Chile through time series techniques: A comparative study. PLOS One, 16(4), e0245414.

Gecili, E., Ziady, A., & Szczesniak, R. D. (2021). Forecasting COVID-19 confirmed cases, deaths and recoveries: Revisiting established time series modeling through novel applications for the USA and Italy. PLOS One, 16(1), e0244173.

Lynch, C. J., & Gore, R. (2021). Short-range forecasting of COVID-19 during early onset at county, health district, and state geographic levels using seven methods: Comparative forecasting study. Journal of Medical Internet Research, 23(3), e24925.

2.2. The use of Median as a measure of central tendency for the metrics seems appropriate. However, this should be supported with relevant references. Recommend adding references to the “Standard metrics” section that discusses the use of median for testing for statistically significant differences across forecasts. A general example and a COVID-context example at different geographic resolution levels are provided:

Lynch, C. J., & Gore, R. (2021). Application of one-, three-, and seven-day forecasts during early onset on the COVID-19 epidemic dataset using moving average, autoregressive, autoregressive moving average, autoregressive integrated moving average, and naïve forecasting methods. Data in Brief, 35, 106759.

Hyndman, R. J., & Athanasopoulos, G. (2018). Forecasting: principles and practice. OTexts.

3) The “Methods” section requires some additional information. Specific points follow:

3.1. With respect to “outlier errors” mentioned in section 1.6 “Standard metrics”, please define how outliers are calculated within the context of this study.

4) The “Abstract” mentions that the provided model both performs better and worse than existing models for various scenarios. The testing metric, i.e., Absolute Error, Weighted Interval Score, etc., should be provided as well along with their corresponding sample sizes and p-values.

5) The “Results” section is underspecified.

5.1. This is a comparative study, and yet, the presentation of results is the shortest section in the article. This section lacks any quantitative support beyond pointing at the figures. Many generic claims are made such as “…shows improvement relative to other baselines…” without providing any statistical support to backup the claim. This section needs to be expanded to provide an in-depth comparison of the forecasting techniques, to include the presentation of tests for significant differences between techniqeus.

6) A “Limitations” section should be added to the text prior to the “Conclusion”. This section should discuss the limitations of the study, the generalizability of the results, and validity concerns.

Additional Comments – In order of appearance:

*) Common issue across many of the figures, text is small and difficult to read and difficult to related points to their corresponding values on the y-axis. Please assess all figures for readability.

Reviewer #2: In this manuscript the authors retrospectively evaluate the use of COVIDici, a mechanistic modeling tool designed to capture the French epidemic with age classes and vaccine interventions, as a longer-term forecasting tool for ICU bed occupancy at the regional and national level. This tool is a non-Markovian discrete-time model which can forecast up to five weeks at local and even departmental levels. The objective here was to determine model performance in anticipating periods of peak ICU overload. The evaluation of COVIDici was carried out by making a comparison to other baseline models, such as ETS+ARIMA, AR-Lasso, and a Naïve auto-regressive model, that also predict ICU bed occupancy in France. The goals of this manuscript are clearly articulated, supported by previous publications, and the authors argue, and I agree, that compartmental model forecasts are valuable. My main concern about the manuscript is the lack of statistical quantification of the differences between the forecasting methods that were compared, and lack of detail about the model fitting in the methods section. Below are major and minor comments.

Major Comments:

1. Although many of the methods used to perform this retrospective evaluation have already been published, the manuscript lacks detail about model evaluation and fitting here in this manuscript. This needs to be addressed with a revision to section 1.3 Calculation. The statement, “Details on the inferred parameters, their prior values and distributions are provided in [16],” is not adequate, especially since the model presented in Figure 1 [16] does not include vaccinated susceptibles. It would be helpful to have a table of mathematical notation and parameters in the supplement describing their definitions and how each of these were fitted, similar to [16]. More detail about the calculation and maximization of the Poisson likelihood to get the expectation and the variance of the infection-to-hospitalization delay using MCMC also needs to be added to the supplemental, as well as a justification for a Poisson model. The algorithms and software used for model evaluation and fitting should also be described and cited.

2. The forecasting methods are qualitatively compared in Figure 3, and the authors use several metrics (e.g. AC, ECR, WIS, Brier score) to quantify forecasting errors to compare models. Comparisons in forecasting error are shown in Figure 4, and the authors claim that COVIDici is a top choice for four-week horizons. The forecasts for each model should be compared statistically (i.e. with measurements of uncertainty such as credible intervals for each curve) in order to determine when COVIDici outperforms other methods. If possible, it would also be valuable if the authors could quantify the expected impacts of the forecasting errors. For example, does the difference in percentage of incorrect forecasts for ICU overload between each model in Figure 5A at four weeks translate into substantial procedural differences in hospitals? Stronger evidence is needed to claim that COVIDici is a better forecasting model over longer time scales, and this should be updated in the Results and Discussion.

3. Inline 263 the authors state, “Figure 5C breaks these metrics down by region which supports similar conclusions.” There is so much overlap in the forecast results by region in 5C, that I disagree that any conclusion can be drawn here. The authors need to further explain and justify this statement.

Minor comments:

4 – “has led to”

Figure 3B – the bold vertical text is hard to impossible to read. This should be fixed.

Figure 3C and Figure 1 in Supplemental should have X- and Y-axes with units.

249 – Please refer to a figure here.

Figure 4A – Why does WIS have a double asterisk in this panel?

Figure 5C – Y-axis needs a label.

Supplemental – In the Calculation section, the authors state, “As a result, we fitted a skewed normal distribution when the point forecast value was greater than 6 (daily events) or a log-normal distribution when the point was less than 6 (i.e. close to zero) using quantile matching.” Please give a reason or citation for why 6 is your cutoff value and why you change the distribution as these values change.

**Have the authors made all data and (if applicable) computational code underlying the findings in their manuscript fully available?**

Reviewer #1: Yes

Reviewer #2: Yes

PLOS authors have the option to publish the peer review history of their article (what does this mean?). If published, this will include your full peer review and any attached files.

Reviewer #1: **Yes: **Christopher Lynch

Reviewer #2: No
---

## [Decision Letter · Decision Letter 1]

29 Mar 2024

Dear Dr. Massey,

Thank you very much for submitting your manuscript "Real-time forecasting of COVID-19-related hospital strain in France using a non-Markovian mechanistic model" for consideration at PLOS Computational Biology. As with all papers reviewed by the journal, your manuscript was reviewed by members of the editorial board and by several independent reviewers. The reviewers appreciated the attention to an important topic. Based on the reviews, we are likely to accept this manuscript for publication, providing that you modify the manuscript according to the review recommendations.

Both reviewers have recommended the manuscript be accepted, but make some minor suggestions for your consideration. Once you have had the opportunity to consider and respond to these suggestions, we should be able to accept the manuscript without further peer review.

Sincerely,

Virginia E. Pitzer, Sc.D.

Section Editor

PLOS Computational Biology

Virginia Pitzer

Section Editor

PLOS Computational Biology

Both reviewers have recommended the manuscript be accepted, but make some minor suggestions for your consideration. Once you have had the opportunity to consider and respond to these suggestions, we should be able to accept the manuscript without further peer review.

Reviewer's Responses to Questions

**Comments to the Authors:**

Reviewer #1: Thank you for taking the prior review comments to heart and making significant changes to your submission. The updates made to the manuscript, particularly with respect to the statistical presentation of results, has greatly improved the overall quality and its readability. The inclusion of Figures 5 and 6 within the main document enhance the transparency of the manuscript and provide greater credibility to the outcomes. Additionally, the incorporation of the limitations section in 3.2 helps to round out the work, provides a solid transition from the Discussion to the Conclusion, and provides a level of self-awareness to the discussion.

Minor suggestion: It may be worthwhile to remind the reader at the start of the Limitations section that these limitations should be interpreted within the perspective that COVIDici was being utilized and updated during the pandemic. Limitation 2 may be a very enlightening point for readers in helping to understand the importance of these types of studies in evaluating the effectiveness of techniques that have been applied in practice during a practical setting to help in informing future practitioners in future global crises.

Reviewer #2: Overall, I think the authors have addressed all the reviewer comments and I have no reservations about moving this manuscript forward for publication. I have several comments below I think the authors should consider to further strengthen their paper.

Major Comment:

The authors clearly state this is retrospective study and should be evaluated in the context of decreased surveillance. These are important points and I appreciate the authors making these clear. However, we are mostly past the pandemic, and it would be helpful for the authors to explain how their analysis can help us prepare for the future. I think it would add impact to the paper if the authors could add information about how these results can more generally inform our ability to make forecasts for hospital strain during disease outbreaks or to prepare for the next pandemic. The only place the authors really do this is in the last sentence in the Author summary, but this could be strengthened. I think statements about the importance of this work to the future need for this type of forecasting should be added to the abstract, the last sentence of the introduction in lines 59-60, and the last paragraph of the conclusion.

Minor comments:

Revise the first sentence of the Author Summary to: “The US and European Covid-19 Forecast Hubs focus on metrics such as deaths, new cases, and hospital admissions, but do not offer measurements of hospital strain like critical care bed occupancy…”

Line 72 – Please provide the doi associated with the Zenodo repository.

Line 150 – Revise the sentence in line 150 to: “The random variable can reasonably be assumed to be Poisson-distributed. This assumption works well for small admission numbers because the population sizes of the investigated locations are large while the daily individual probability of being admitted in an ICU for COVID-19 is small.”

**Have the authors made all data and (if applicable) computational code underlying the findings in their manuscript fully available?**

Reviewer #1: Yes

Reviewer #2: Yes

PLOS authors have the option to publish the peer review history of their article (what does this mean?). If published, this will include your full peer review and any attached files.

Reviewer #1: **Yes: **Christopher J. Lynch

Reviewer #2: No

Figure Files:

Data Requirements:

Reproducibility:

References:

---

## [Editor Report · Decision Letter 2]

1 May 2024

Dear Dr. Massey,

We are pleased to inform you that your manuscript 'Real-time forecasting of COVID-19-related hospital strain in France using a non-Markovian mechanistic model' has been provisionally accepted for publication in PLOS Computational Biology.

Best regards,

Virginia E. Pitzer, Sc.D.

Section Editor

PLOS Computational Biology

Virginia Pitzer

Section Editor

PLOS Computational Biology

---

## [Editor Report · Acceptance letter]

14 May 2024

PCOMPBIOL-D-23-01056R2 

Real-time forecasting of COVID-19-related hospital strain in France using a non-Markovian mechanistic model

Dear Dr Massey,

I am pleased to inform you that your manuscript has been formally accepted for publication in PLOS Computational Biology. Your manuscript is now with our production department and you will be notified of the publication date in due course.

With kind regards,

Anita Estes
